# Stress-induced plasticity of dynamic collagen networks

Jihan Kim[1], Jingchen Feng[2], Christopher A.R. Jones[1], Xiaoming Mao[3], Leonard M. Sander[4], Herbert Levine[2,5] & Bo Sun [1]

The structure and mechanics of tissues is constantly perturbed by endogenous forces originated from cells, and at the same time regulate many important cellular functions such as migration, differentiation, and growth. Here we show that 3D collagen gels, major components of connective tissues and extracellular matrix (ECM), are significantly and irreversibly remodeled by cellular traction forces, as well as by macroscopic strains. To understand this ECM plasticity, we develop a computational model that takes into account the sliding and merging of ECM fibers. We have confirmed the model predictions with experiment. Our results suggest the profound impacts of cellular traction forces on their host ECM during development and cancer progression, and suggest indirect mechanical channels of cell-cell communications in 3D fibrous matrices.

[1] Department of Physics, Oregon State University, 301 Weniger Hall, Corvallis, OR 97331-6507, USA. [2] Center for Theoretical Biological Physics, Rice University, 6100 Main Street, Houston, TX 77005-1892, USA. [3] Department of Physics, University of Michigan, 450 Church St, Ann Arbor, MI 48109-1120, USA. [4] Physics and Complex Systems, University of Michigan, Ann Arbor, MI 48109-1120, USA. [5] Department of Bioengineering, Rice University, Houston, TX 77030-1402, USA. Jihan Kim and Jingchen Feng contributed equally to this work. Correspondence and requests for materials should be addressed to H.L. (email: herbert.levine@rice.edu) or to B.S. (email: sunb@onid.orst.edu)

Our tissue is continually changing. At a fundamental level, the main scaffold of our connective tissue, a matrix of collagen fibers, is constantly remodeled by the cells living therein[1, 2]. Either as a regular process of tissue homeostasis, or as a response to inflammation and wounding, new collagen fibers are synthesized[3], while existing fibers are degraded[4]. These biochemical interactions between the cells and the collagen matrix are crucial to maintain the integrity of our living tissue[5].

Cells also interact with the collagen matrix physically[6] and probe the nonlinear elasticity[7, 8] and viscoelasticity[9, 10] of the extracellular matrix (ECM). The physical interactions, such as the pushing forces from membrane protrusions and the pulling forces from cell contraction, are generally considered to be small and to reversibly deform the surrounding matrix. For example, a fundamental assumption of the widely employed three dimensional (3D) traction force microscopy is that once cell-generated forces are released, the matrix will bounce back to its non-stressed configuration[11, 12].

Remarkably, recent experiments have reported densified, aligned collagen fibers between clusters of cancer cells[13, 14]. These observations suggest that collective contraction between cell clusters may cause large deformations in the ECM. It is questionable, therefore, if the assumptions of small and reversible deformations still hold in the case of well-dispersed cells. In fact, other workers have demonstrated the inelastic behaviors of collagen networks[15, 16] and irreversible alignment of ECM fibers near single contracting cells[17].

In this paper, we find a pair of breast cancer cells alone can increase the local fiber density of reconstituted collagen matrices by more than 150%. We will show that these large deformations are irreversible, history dependent, and significantly change the

ECM micromechanics. The observed ECM remodeling is purely mechanical, without the creation or degradation of fibers. To provide a fundamental understanding of ECM plasticity, we have devised a computational model, based on the dynamics of crosslinks and fiber entanglement. Instead of taking a mean-field approach similar to that of Nam et al.[18], we fully capture the complex disordered network structure. Because biopolymer networks are strongly disordered networks and the disorder is important in many important phenomena. Our model gives agreement with cellular experiments, elucidates microscopic details of force and energy distributions in disordered plastic networks, and predicts rich bulk rheology beyond the widely accepted viscoelasticity of a collagen matrix[19–21]. We have validated these predictions with experiments, and suggest a mechanical mechanism that contributes to a dynamic, reconfigurable ECM without the need for chemical modifications.

## Results

**Cell traction forces induce plastic ECM deformations.** We have studied the time-dependent 3D ECM remodeling by cell pairs with quantitative imaging. Figure 1a, b shows an example where two breast cancer cells (MDA-MB-231) are embedded in a type-I collagen gel (see also Supplementary Movies 1, 2). Immediately after the gelation process completes, the cells start to generate traction forces which deform the local ECM, while the morphologies of the cells remain rounded. Using confocal reflectance imaging, we find that the matrix microstructure is most significantly remodeled between the cells, a region we will refer to as a collagen bundle. A collagen bundle consists of aligned, and densified collagen fibers (Fig. 1b) connecting the cell pair. We find that the formation of collagen bundles is robust,

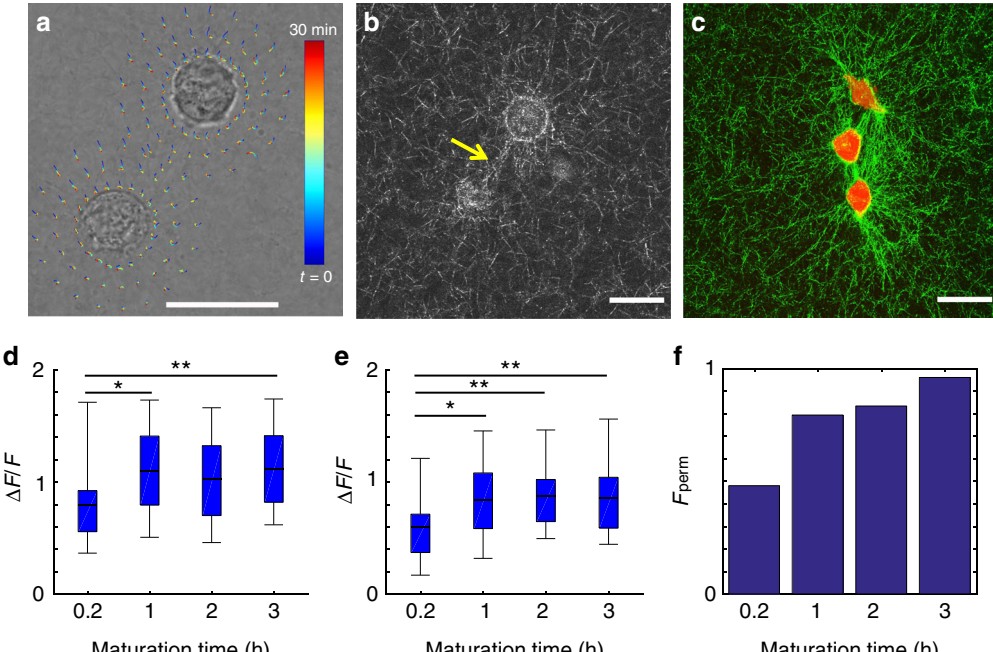

**Fig. 1** Cell traction forces irreversibly induce the formation of collagen bundles. **a** Reconstructed streamlines showing the spatial-temporal profile of the cell-induced matrix deformation. The deformation field from frame to frame is calculated via reflectance particle image velocimetry[22]. Color code (blue to red) is linearly proportional to maturation time 0–30 min). **b** Confocal reflection image of the collagen matrix showing a collagen bundle (arrow) between two MDA-MB-231 cells. **c** Collagen bundles simultaneously form between multiple cell pairs. Red: GFP-labeled MDA-MB-231 cells. Green: reflectance image of collagen fibers. **d** The relative reflectance intensity $\Delta F/F$ of collagen bundles compared to the background. **e** $\Delta F/F$ of collagen bundles after disrupting the cell traction forces by Cytochalasin-D treatment. In **d**, **e**, ~30 cell pairs are sampled for each maturation time. The thick black lines, box edges and whiskers represent the median, first/third quartiles, and lower/upper 5% values, respectively. ANOVA and Fisher's least significant difference procedure is used to evaluate the difference of $\Delta F/F$ corresponding to different maturation times. *$p < 0.05$, **$p < 0.01$. Differences between non-labeled pairs are not significant. **f** Fraction of permanent collagen bundles $F_{perm}$ as a function of maturation time. Scale bar: 20 μm

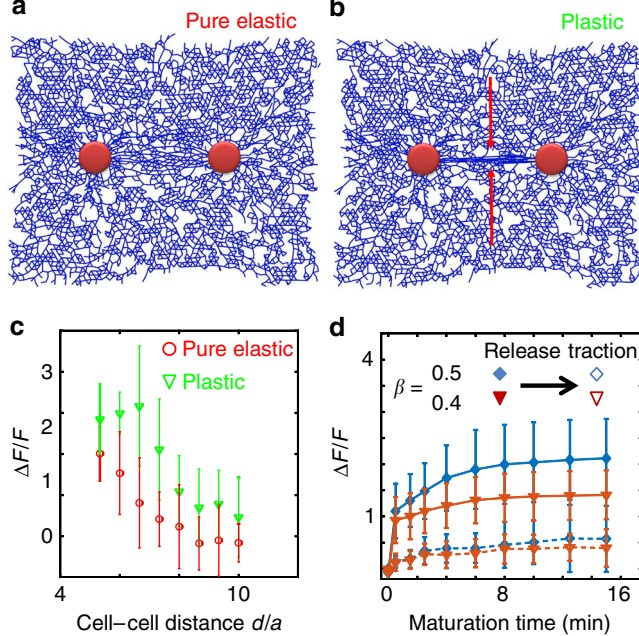

**Fig. 2** Simulation of collagen bundle formation by contracting cell pairs. **a** The network configuration in an elastic model (without any sliding or merging events). **b** The network configuration predicted by our plastic model. **c** The relative increase of fiber density $\Delta F/F$ of collagen bundles compared with the background at varying cell–cell distance. Here the cell–cell distance $\alpha$ is normalized by the cell size $\alpha$. Red: plastic deformation with sliding events. Blue: pure elastic response of the network. At any given distance, the results from elastic ($T = 0$ min) and plastic ($T = 15$ min) are statistically distinct ($t$-test, $p = 0.0007$, $N = 8$). **d** The irreversibility of a collagen bundle depends on both cell contractility ($\beta$) and maturation time $T_d$. Here the irreversibility is characterized by $\Delta F/F$ after the cell traction force is released. The cell–cell distance is fixed at $d = 7a$. Error bars in **c**, **d**: mean ± SD, obtained from eight different realizations

and generally present between all cell pairs which are within 80 μm distance. (Fig. 1c, Supplementary Methods, and Supplementary Figs. 1, 2).

In order to quantify the level of ECM remodeling, we characterize the collagen bundles by their relative reflectance intensity with respect to the matrix far from the bundles $\Delta F/F$ (Methods section). Most collagen bundles are 50–150% brighter than the average background intensity, and are significantly brighter than the background fluctuation ($\Delta F/F \geq 2\frac{\delta F}{F}$, where $\delta F$ is the standard deviation of the background intensity as detailed in Supplementary Methods and Supplementary Fig. 1. See also Supplementary Fig. 3 for the characterizations of fiber alignment). Therefore the collagen bundles are distinct from the naturally occurring density fluctuations of the collagen gel[23].

We find that the relative intensity of a bundle increases with its maturation time, which we count from when the collagen gel is formed. Both single bundle continuous imaging and the statistics of snapshots of multiple bundles confirm that $\Delta F/F$ reaches a plateau after an hour (Fig. 1d).

A common assumption in cell mechanics is that once a traction force is released, the matrix will relax to its original stress-free state. Indeed, when the cell traction forces are released by Cytochalasin-D treatment, the relative intensity of the collagen bundles decreases, particularly for those with short maturation times. However, removing the mechanical stress does not fully remove the collagen bundles. Instead, a significant amount of residual strain remains in the regions of collagen bundles (Fig. 1e, Supplementary Fig. 4). We consider a collagen bundle to be a

permanent one if its relative intensity is significantly higher than the background ($\Delta F/F > 3\frac{\delta F}{F}$) even after treating the cells with Cytochalasin-D. By counting more than 80 cell pairs, we find that the fraction of permanent collagen bundles, indicative of the plasticity of ECM deformations, increases with maturation time (Fig. 1f) to nearly unity.

To further demonstrate the mechanical origin of collagen bundles, we have developed a microstretching device, which generates local mechanical deformation in 3D collagen matrix similar to a cell pair (Supplementary Methods, Supplementary Fig. 5, and Supplementary Movies 3, 4). When extensional stress is applied for a short period of time, the matrix will almost fully recover to its original configuration. When the dwell time is increased, regions of densified fibers persist even after the stress is released. These observations suggest that formation of collagen bundles and the history-dependent plastic ECM remodeling have a purely mechanical origin. Indeed, we find extensive bundle formation even after inhibiting matrix metalloproteinase (Supplementary Methods and Supplementary Fig. 6).

**Computational modeling of ECM plasticity**. We hypothesize that the observed plasticity of the collagen matrix is a result of the irreversible dynamics of cross-links and fiber entanglement. To test the hypothesis, we have developed a computational model based on a diluted triangular lattice. We treat the collagen matrix as an athermal network of fibers that resist bending and stretching[24–27]. Using experimentally derived stretching and bending moduli of fibers[28], and the coordination number of the network[29], we construct a minimal representation of the matrix. The linear and nonlinear elasticity of this model has been extensively studied. It is known that in the linear regime, the elastic energy is dominated by bending energy of the fibers, because the network has less connectivity than at the central-force isostatic point[25, 26]. As the model is deformed beyond the linear elasticity regime, the elastic energy becomes stretching dominated, and the shear modulus increases by more than an order of magnitude (strain-stiffening), in good agreement with observations of various biopolymer gels[27, 28, 30–32].

The relationship between this two-dimensional (2D) model and a real 3D biopolymer networks has been examined in several recent papers[25, 33, 34]. In particular, in ref. [34] lattice models are compared to off-lattice versions. The result of all these studies is that the main features of the elastic behavior are faithfully captured, though there may be differences in details, particularly in the nonlinear regime.

In contrast to previous models that assume static network connections, we consider the cross-links between fibers to be dynamic: when the force loaded on a cross-link exceeds a threshold, two fibers will have a probability to detach and reconnect to lower the elastic energy, or a branching fiber has a probability to peel apart further at the branching point, which we call sliding. In addition, we consider the merging of adjacent fibers within a critical distance, which can be either due to fiber entanglement or chemical bond formation. Both sliding events and merging events are intrinsically irreversible and contribute to the plasticity of the model network (Supplementary Methods and Supplementary Fig. 7).

We first test if the model reproduces the observed properties of our collagen bundles. Because experiments observe mostly rounded cells (Supplementary Fig. 2), we model contractive cells as circles embedded in the network and isotropically shorten all fibers inside the circle by the fraction $\beta$. The contracted cell size $\alpha$ in proportion to the matrix pore size has been chosen to be consistent with experimental measured cell radius ($17.2 \pm 2.6$ μm) and pore size ($3.0 \pm 0.7$ μm).

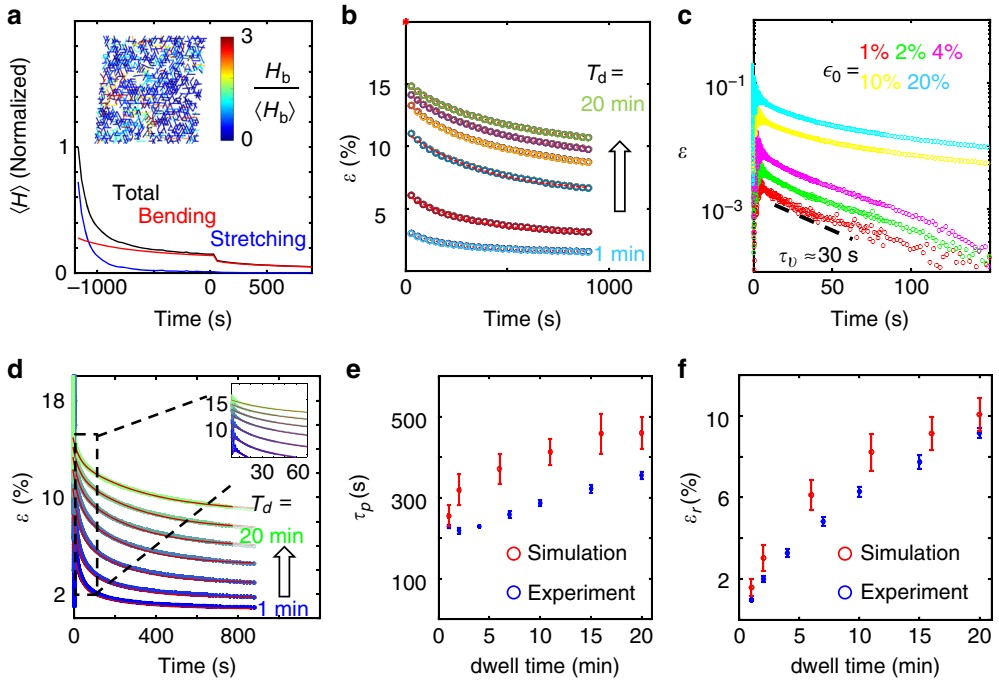

**Fig. 3** Bulk relaxation kinetics of collagen matrices. **a** The normalized elastic energy per fiber $\langle H \rangle$ over the course of relaxation of a model network. Black: sum of bending and stretching energy. Red: bending energy. Blue: stretching energy. All three curves are normalized by the total energy per fiber at $t = -20$ min. The network is sheared to 20% at $t = -20$ min, and released at $t = 0$. Inset: The network configuration after 20 min of relaxation ($t = 1200$ s). The fibers are color-coded according to the bending energy per unit length of each fiber $H_b$, normalized by the ensemble average $\langle H_b \rangle$. **b** Simulated strain decay kinetics with 20% initial strain and varying dwell times $T_d = 1, 2, 6, 10, 16$, and 20 min. The dashed lines are fits to a single exponential. **c** Experiments show strain relaxation kinetics $\varepsilon(t) - \varepsilon(\infty)$ depend on the initial strain, and at small initial strains, the relaxation follows a single exponential function. Here $\varepsilon(\infty)$ is approximated by the strain measured after 15 min of relaxation, Supplementary Fig. 16 for results with extended relaxation time. **d** Experiments show strain relaxation kinetics depends on the dwell time $T_d$. Colors of the symbols (blue to green) correspond to the increasing dwell time of 1, 2, 4, 7, 10, 15, and 20 min. Red lines are fit to double-exponential functions $\varepsilon(t) = a \exp(-t/\tau_v) + b \exp(-t/\tau_p) + \varepsilon_r$. Here $\tau_v$ is independent of dwell time $T_d$, $\tau_p$, and $\varepsilon_r$ are allowed to vary with $T_d$. Inset: zoom-in to the initial phase of the relaxation. **e** The plastic time scale $\tau_p$ as a function of dwell time $T_m$. **f** The residual strain $\varepsilon_r$ as a function of dwell time $T_d$. Error bars in **e**, **f** are means and standard deviations from eight different samples

At $T = 0$ (immediately after cell contraction), the network configuration is determined by minimizing the elastic energy and no sliding or merging events are allowed to occur (Fig. 2a). Every half-minute thereafter, we allow all possible sliding events to occur deterministically and all merging events to occur with probability $P_{\text{merging}}$. This approach is based on the assumption that the time scale of sliding events is much faster than that of merging events (Supplementary Methods). After roughly 15 min of maturation time, sliding causes the fibers to continuously flow into the central region between the cells, as we observe in the formation of collagen bundles (Fig. 2b).

To quantitatively compare the simulation results and experiments, we calculate $\Delta F/F$ by measuring the relative increase of fiber density in the bundle region. As shown in Fig. 2c, sliding events significantly increase the fiber density in the bundle as compared with purely elastic deformations. On the other hand, the fiber alignment, as characterized by the nematic order, is not sensitive to the sliding events (Supplementary Fig. 8). Simulation shows that $\Delta F/F$ decreases with cell–cell distance, which is also consistent with the experiment (Supplementary Fig. 2 and also Supplementary Fig. 9 for elongated cell shapes).

Note that our computational cells contract by the large fraction $\beta$, whereas in our experiment (and in many others) the observed cell area does not change much in the process. This paradox is only apparent: we are not modeling the plasma membrane of the cell (which determines the observed area) but rather the motion of the points where the cell is attached to the surface. These points do contract strongly due to the action of motor proteins inside the cells. Further, we use a continuous disk of

attachment rather than discrete points. This is also not a real problem: we are interested in deformations of the matrix at distances large compared to the spacing between adhesions. There is no difference between discrete and continuous adhesion in this regime. Another indication of this fact is that our results our do not change much for elongated cells (Supplementary Fig. 9).

Our model also allows us to systematically examine the irreversibility of the collagen bundles. To this end, we have varied the contractility ($\beta$) and maturation time $T_m$, and measured the density increase in the bundle region as compared to the average background density $\frac{\Delta F}{F}$ after cell traction is released. As shown in Fig. 2d, the irreversibility of ECM builds up as a function of maturation time $T_m$. Intriguingly, the sliding events and merging events play separate roles. The former mainly contributes to the enhanced fiber concentration before releasing cell traction forces, and the later mainly contributes to the irreversibility of collagen bundle formation (Supplementary Fig. 8).

**Microscopic reconfigurability impacts the bulk mechanics.** Although the collagen bundles are localized structural features in the fibrous network, we expect their mechanistic origin, namely the sliding and merging events may have a profound impact on the bulk properties of the collagen matrix. To examine this effect, we studied the history-dependence of the relaxation dynamics of the model networks under macroscopic shear deformation. In particular, we held the matrix at an initial shear strain of 20% for a dwell time of $T_d$ to allow plastic reconfiguration. We then

released the boundary stress and monitored the strain relaxation as a function of time $\varepsilon(t)$.

Once the network is released the shear strain drops from $\varepsilon_0 = 20\%$ to a non-zero value $\varepsilon(0^+)$ due to purely elastic relaxation. Because we do not consider viscosity effects in our model, this initial drop happens instantaneously. In a real collagen matrix, viscoelasticity due to the collagen-solution interaction and filament entanglement necessarily exist. However it is known that the viscoelasticity time scale is below 1 min and much shorter than the plasticity time scale we discuss here[35, 36]. Thus we ignore viscoelasticity in our modeling and only focus on plastic events including sliding and merging.

Interestingly, we find that during the dwell time the stretching energy decays much faster than the bending energy, implying transitions from stretching-dominated states to bending-dominated states through sliding events. After 20 min of dwell time, the network is dominated by bending energy, and approaches a finite value asymptotically (Fig. 3a). A snapshot of the network after 20 min of relaxation suggests a highly heterogeneous distribution of bending energy after the system is relaxed plastically (Fig. 3a inset). We find the subsequent decay of strain follows a single exponential function $\varepsilon(t) = (\varepsilon(0^+) - \varepsilon_r)\exp(-t/\tau_p) + \varepsilon_r$ for $t > 0$ and that the decay is slower with increasing dwell time, $T_d$, (Fig. 3b). Because longer dwell time, $T_d$, allows the network to reduce the number of high-stress bonds through sliding events, we expect a negative correlation between $T_d$ and $1/\tau_p$, the rate of plastic relaxation. Indeed, we find that both $\tau_p$ and the asymptotic residual strain $\varepsilon_r$ increase with the dwell time (see also Supplementary Figs. 10–12). Together, these model results predict that the collagen matrix may exhibit history-dependent strain relaxation, and that the relaxed state is a permanent reconfiguration of the original matrix.

We have confirmed the above theoretical predictions by studying the shear strain relaxation kinetics of a collagen matrix using a parallel plate rheometer. Since both viscoelastic and plastic dynamics are present, we begin by studying the relaxation from small initial strains. In this case stress is also small and we expect few sliding events and mostly viscoelastic relaxation. Indeed, we find that up to $\varepsilon_0 < 5\%$, the strain kinetics can be well characterized by a single exponential function with a time scale $\tau_v \approx 30$ s, presumably determined by viscoelasticity of the matrix (Fig. 3c). However, when the initial strain approaches a threshold ($\approx 10\%$, Supplementary Fig. 13) of linear elasticity, or beyond, a single exponential is no longer sufficient. Under such conditions, we expect the relaxation to be dominated by viscoelasticity at short time scales and plasticity at longer time scales.

Indeed, as shown in Fig. 3d, when collagen matrices relax from 20% initial strain, the relaxation kinetics fit well with double-exponential functions $\varepsilon(t) = a\exp(-t/\tau_v) + b\exp(-t/\tau_p) + \varepsilon_r$. Here $\tau_v = 29.6$ s is independent of the dwell time $T_d$ (Supplementary Fig. 14), and matches well with the viscoelastic time scale obtained from small strain relaxation kinetics in Fig. 3c. Consistent with our model assumption, $\tau_p$ is well separated from $\tau_v$ by an order of magnitude. In addition, we have confirmed that both $\tau_p$ and $\varepsilon_r$ increase monotonically with longer dwell time $T_d$ (Fig. 3e, f), as predicted by the model (see also Supplementary Figs. 15–17 for additional tests with collagen and fibrin gels).

**Cell-induced plastic deformations remodel the ECM micromechanics.** After demonstrating the effects of microscopic plasticity on the structural remodeling of collagen ECM at both cellular and macroscopic scales, we have also examined the accompanying changes in the ECM mechanical properties. We first study the micromechanical signatures of collagen bundles created by cell pairs[37]. To this end, we have embedded probe

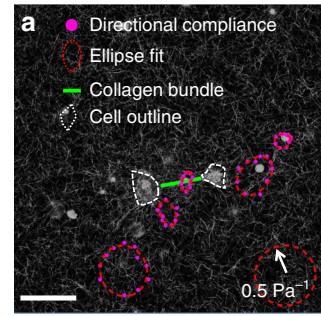
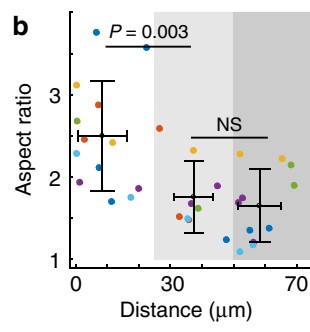

**Fig. 4** The micromechanics of collagen ECM in the vicinity of cell-induced collagen bundles after traction forces are released. **a** The confocal reflection image and directional compliance given by five probe particles around a collagen bundle in a typical experiment. The compliance is scaled linearly into real space such that an isotropic response of 0.5 Pa$^{-1}$ would be plotted as a ring with the size of the bottom right circle. Magenta dots: experimentally measured directional compliance. Red circles: the compliance ellipse, i.e., the elliptical fit to the magenta dots. White dashed lines: outlines of MDA-MB-231 cells after Cytocytochalasin-D treatment. Green line: the location of collagen bundle. Scale bar: 50 μm. **b** The aspect ratios of the compliance ellipses at varying particle-to-bundle distances $d$. Symbols of different colors correspond to results measured around different bundles. We divide all the data into three groups $d < 25\,\mu m$, $25 \le d < 50\,\mu m$, and $50 \le d < 75\,\mu m$. Error bars represent the mean and standard deviations of each group. ANOVA analysis shows that the aspect ratios close to the collagen bundles ($d < 25\,\mu m$) are significantly higher than the values further away

microparticles in collagen matrices together with MDA-MB-231 cells. After more than 3 h of maturation time, we released the cell traction forces with Cytocytochalasin-D, leaving only the plastic deformations. Using holographic optical tweezers[38], we measured the directional compliance $J(\theta)$ from probe particles within 80 μm of collagen bundles. Here $\theta$ represents the direction along which small optical forces ($\sim pN$) are applied. $J(\theta)$ is defined as $J(\theta) = 6\pi a \frac{\Delta d_\theta}{F_\theta}$, where $a$ is the particle radius, $\Delta d_\theta$ and $F_\theta$ are the particle displacement and optical force in the $\theta$ direction respectively (see also Supplementary Fig. 18). For linear elastic materials, $J(\theta)$ equals to the elastic compliance.

The characteristics of $J(\theta)$ show that the presence of collagen bundles significantly contributes to the micromechanical heterogenity in the ECM. Figure 4a shows typical measurements around a collagen bundle (green line) between two MDA-MB-231 cells (white outlines). For each of the five probe particles, we measured the directional compliance at 30° increments in $\theta$, and the resulted directional compliance $J(\theta)$ was fitted with an ellipse (compliance ellipse) using $J$ as a polar distance. The aspect ratio of the compliance ellipse quantifies the local mechanical anisotropy.

To better visualize the spatial pattern of the micromechanics, in Fig. 4a we overlaid the confocal reflection image with the measured directional compliance (magenta dots) and their elliptical fits (red dashed lines). The compliance is scaled linearly into a closed curve centered around each probe particle (the scaling factor is indicated by the bottom right circle). We find the particle on the collagen bundle gives a highly an isotropic local compliance, with approximately twice more compliance in the direction perpendicular to the bundle than parallel. This is expected because collagen bundles consist of aligned fibers whose bending elasticity is softer compared with stretching. Moving away from collagen bundle, the micromechanical compliance becomes increasingly isotropic. This is evident from Fig. 4a, and is also confirmed by sampling multiple bundles. Figure 4b shows

the aspect ratios of the compliance ellipses at various particle-to-bundle distances $d$. Close to collagen bundles ($d < 25\,\mu m$), the aspect ratio is significantly higher than the values measured further away ($25 \le d < 50\,\mu m$ and $50 \le d < 75\,\mu m$) from the bundles.

## Discussion

We have demonstrated that traction forces from cell pairs are capable of locally remodeling 3D collagen ECM into densified, aligned fiber bundles. Rather than being small perturbations to the ECM, as typically assumed for the cell traction forces, fiber density in the bundle region increases dramatically (by as much as 150%), which is comparable with previous observations of ECM remodeling by clusters of cells[13, 14]. The micromechanics of the ECM is also significantly modified, with greater mechanical anisotropy close to the collagen bundles. These results suggest that collagen ECM is highly susceptible to mechanical remodeling by the cells.

While the formation of collagen bundles would occur for reversible elastic deformations, either linear or nonlinear[31], we find that collagen bundles persist even after cell traction forces are removed. Therefore the collagen bundles are cell-induced permanent deformations of the ECM, which is only possible if the collagen matrix is plastic. To understand the implications of ECM plasticity, we devised a computational model based on irreversible sliding and merging of fibers under stress in a model network. Our model not only reproduces the irreversible structural remodeling by cell traction forces, but also agrees with bulk rheological measurements on collagen gels.

While sliding and merging events produce good agreement with experiment, we could also consider other sources of ECM plasticity. Each collagen fiber consists of several weakly bound parallel fibrils. Stretching of fibers causes sliding between fibrils, which permanently lengthen the fiber. Intrafibrillar sliding has been shown to contribute the history-dependent elasticity of collagen gels, particularly when the gels are probed under repeated stress-relaxation cycles[39]. Although fiber lengthening is likely to occur in our experiments, it does not explain the densified and aligned collagen bundles between cells, nor would it lead to residual strains after bulk shearing. However, the collagen matrix used in our study is a network of fibers that interact non-covalently. Weak interactions, such as hydrogen bonds and electrostatic interactions allow force-dependent unbinding and rebinding between collagen fibers[18], which is similar to the sliding events we have proposed here. These dynamic bonds have been shown to contribute to the plasticity of collagen matrix in vitro, as well as for isolated mouse tissues[17]. Interestingly, while it was found that higher strain magnitude leads to faster stress relaxation in collagen matrix[18], we show that the strain relaxation is slowed down by longer dwell time. This apparent contrast highlights the complex strain-stress relation of collagen matrices, a very direct consequence of plasticity.

As the major component of connective tissues, and a semi-flexible, subisostatic polymer network, the collagen matrix demonstrates nonlinear elasticity which can be controlled by external stress or strain[27]. This mechanical reconfigurability is further expanded by the stress-activated plasticity reported here. We expect future studies will take advantage of these effects to establish collagen matrix as a mechanically programmable material which has excellent biocompatibility[40, 41]. The plasticity of collagen matrix also implies a new mode of 3D cell-cell interaction in tissues: the collagen bundle from a pair of cells poses microstructural guidance to nearby cells through contact guidance[42–44]; and at the same time creates micromechanical guidance to nearby cells through durotaxis[45, 46]. Such interactions are non-local and long-lasting, and we expect them to have direct impact on the multicellular dynamics in various physiological processes such as cancer metastasis, wound healing and embryo development[47].

## Methods

**Sample preparation and imaging.** Cell-embedded collagen gels are prepared by diluting and neutralizing high concentration type-I collagen solution (10 mg/ml, Corning) with NaOH, cell suspension, growth medium, and 10X PBS into 1.5 mg/ml. The neutralized solution is immediately placed in a tissue culture incubator (NuAire) to polymerize at 37 °C for 40 min, then the maturation time starts to count.

To image the fluorescently labeled MDA-MB-231 cells cultured in collagen gel, we use a laser point-scanning confocal microscope (Leica SPE) equipped with an stage-top incubator (ibidi). Both fluorescent and reflection channels are imaged with either 20X or 40X oil immersion lenses as described previously[22]. To image collagen bundles of various maturation times, the samples are placed in the tissue culture incubator until the time to image. It usually takes less than 10 min to locate the collagen bundles under the microscope. Therefore there is an added uncertainty of ~10 min in the maturation times in the plots of Fig. 1.

To release cell traction force, we dilute Cytochalasin-D (Sigma-Aldrich) with PBS to a 1:1000 ratio and add directly to the 3D culture samples. We allow 2 h to complete the treatment before washing the sample with growth medium.

**Confocal image analysis.** All confocal images are analyzed using NIH ImageJ and homemade Matlab scripts. More detailed procedures are described in Supplementary Methods.

**Measurement and fitting of the bulk relaxation kinetics.** To measure the bulk relaxation kinetics of the collagen gel, we prepare the gel between the two parallel plates of a AR-G2 stress-controlled rheometer (TA Instruments) at 37 °C and concentration of 1.5 mg/ml. Liquid seal and Peltier chamber are used to ensure the gelation condition and prevent evaporation. The plates are stainless-steel, and surface treated with CellTak (Corning) to ensure binding to the collagen. To measure the dependence of strain relaxation on dwell time, the dwell time starts from 1 min and gradually increases to 20 min for each given sample. The initial strains are applied by shearing the sample at 1% per second, until reaching the desired strain magnitudes. For each dwell time, we allow 15 min of relaxation before bringing back the sample to 20% strain. To account for the cumulative effect of the residual strains, we re-calibrate the system after each relaxation cycle by setting the previously relaxed configurations as strain zero states. We have also tested for each dwell time using different samples (Supplementary Fig. 15).

To fit the relaxation kinetics, we use the Matlab nonlinear curve fitting package. More details are given in Supplementary Methods.

**Computation.** The total energy is minimized by using a conjugate gradient algorithm. The parameters for the triangular lattice-based model are: bond occupation probability $p = 0.60$, network size $S = 60 \times 60$ for cell experiments and $S = 40 \times 40$ for bulk rheology, bending stiffness $\kappa = 10^{-3}$ and stretching stiffness $k = 1$ unless stated otherwise. The fiber concentration of collagen bundles $F$ is equal to the total number of bonds in region of bundle (ROB) divided by ROB area. $F = 3p/2S$, where $S = \sqrt{3}/4$ is the area of an undeformed triangle unit. The relative intensity increase is defined as $\Delta F/F = (F_b - F)/F$. The network has free boundary conditions for simulations of cell experiments. The bonds connected to the cell surface are capable of freely sliding along the cell surface. For bulk rheology, the network has fixed boundary conditions on the top and bottom and periodic boundary conditions on the left and right at 20% deformed state. The top boundary condition is changed to zero-stress after releasing the network. For details of sliding and merging events, Supplementary Methods. The simulation codes written as MATLAB scripts can be obtained from the authors upon reasonable request.

**Data availability.** Confocal images not included in the manuscript can be found at: https://doi.org/10.6084/m9.figshare.5279956.v1. All other data are available from the authors upon reasonable request.

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

## Acknowledgements

We are thankful to Prof. Skip Rochefort and Prof. David McIntyre for accesses to experimental equipment. J.K. and C.A.R.J. are partially supported by the National Science Foundation Grant PHY-1400968. J.F. and H.L. are supported by the National Science Foundation Center for Theoretical Biological Physics (Grant PHY-1427654). X.M. is supported by the National Science Foundation Grant NSF-DMR-1609051. H.L. is also supported in part by the Cancer Prevention and Research Institute of Texas Scholar Program of the State of Texas at Rice University.

## Author contributions

H.L. and B.S. initially perceived and designed the study. J.K., C.A.R.J., and B.S. conducted the experimental research. J.F., X.M., L.M.S., and H.L. conducted the theoretical research. J.K. and J.F. made equal contributions to this study. All authors analyzed the data and wrote the manuscript. The authors declare no competing financial interests.

## Additional information

**Competing interests:** The authors declare no competing financial interests.

