## [Peer Review File · Nature Communications]

Reviewers' comments:

Reviewer #1 (Remarks to the Author):

In the paper titled "Stress-induced Plasticity of Dynamic Biopolymer Networks", Kim et al. have performed an experimental and computational study of Biopolymer networks found in the extracellular matrix. The experiments have been performed on Type 1 collagen gels whereas the simulations have been performed on the lattice based (2D and 3D) network models. Based on their experiments, they claim that the network is significantly remodeled by cellular traction forces. The remodeling is irreversible and can occur under the application of microscopic strains also. They have then developed a computational model, which takes into account the remodeling of network by including sliding and merging of network fibers. The model calculations are in qualitative agreement with experiments. The study highlights that traction forces from cell pairs can cause significant alignment of collagen fibers, resulting in fiber bundles. The authors emphasize that the traction forces cannot be treated as small perturbations as they can increase the fiber density by as much as 150%.

The findings of this study are interesting. That collagen matrix is irreversibly remodeled in presence of traction forces or macroscopic strains has been reported in previous experimental studies, the study of Kim et al. provides a computational approach which I believe is well suited to the study of remodeling of ECM. I find the article well written and interesting to warrant a publication in Nature communications. There are, however, certain points that must be addressed by the authors before I can recommend publication.

1. The remodeling here focuses on collagen networks where there is no generation of collagen fibers. The remodeling is purely due to the forces with no change in the total line density. Since remodeling of collagen networks, on a broader perspective, also takes into account generation of fibers, I suggest the authors to clearly mention the absence of generation of collagen fibers in the introduction to the article.

2. The experimental results in Figure 1 are strongly indicative of the remodeling by traction forces. However, the computational model details are not sufficient (including those in the Supplementary information) and occasionally misleading.

a. The authors mention time $T_{\text{sliding}} = 0.5$ min and $T_{\text{merging}} = 25$ min. The only reason for this choice of parameters as mentioned by the authors in the Supplementary information is 'To reach good correspondence with experiments'. What is meant by this? In the computational model, where one minimizes the elastic energy (before sliding/merging events), using any appropriate algorithm (be it conjugate gradient), there is no notion of time. Even if the authors used steepest descent method and other standard assumptions (no hydrodynamics, over-damped), the natural time scale in the system will need to be expressed using viscosity of the underlying solvent and elastic forces in the network. Hence, I do not understand the meaning of T_{sliding} and T_{merging} numerical-values provided. Is it not so that in the simulations, the merging events are 5-times rarer in time than the sliding events (purely based on parameter choice of authors)? Please clarify the concept of time-evolution in the simulations.

b. When sliding events are considered, the authors consider a frozen network configuration, i.e., all those fibers that can slide are allowed to do so, or one follows a priority list in order of most stressed to least stressed with network relaxation between every subsequent sliding event (a quasi-static approach)?

3. Assuming that the authors can explain the choice of $T_{\text{sliding}} = 0.5$ minutes, is this time scale independent of the initial stress/magnitude of traction forces?

4. Page 3, 4th paragraph: "Affine to non-affine transformation through sliding events": Do the authors apply a step-shear to the system? Is the dwell time corresponding to the time for which

the strain is kept constant? The authors say that for a dwell time of 20 minutes, they observe an affine to nonaffine transition that is attributed to plastic events. However, in the reference [21], the nonaffine transition has nothing to do with plastic events. There (in ref 21), the nonaffine transitions are found in a topologically fixed network.

5. In Fig 3D, a double exponential function is used to fit the data. The time scales of the two exponentials are well separated. However in the plot 3D, the time (x axis) is shown in 100s of seconds such that one cannot possibly see the τ_v relaxation at all. The authors should zoom in the short time scale to show the fit to the τ_v .

6. I am really curious about why one should attempt an exponential (or double exponential) fit? Is there an obvious reason to assume exponential relaxation?

A General remark:

The article is a good piece of work. I do not think that it needs to make speculations (last paragraph of discussions section) like "...will take advantage of collagen matrix as a mechanically programmable material which has excellent biocompatibility". I find it highly unnecessary but leave it to authors to decide whether they do anything about it. Similarly the very last sentence of the discussion section: "...cancer-metastasis, wound healing and embryo development." Is this really needed?

Citation error: 23 and 32 are repeated.

Reviewer #2 (Remarks to the Author):

This manuscript reports a combined experimental and computational study of plasticity in fibrous extracellular matrix networks. This topic is currently gaining increasing interest in the context of cancer biology and regenerative medicine, where cell-driven remodeling of the matrix is a central phenomenon.

The main findings of the paper (large deformations applied by cells or external stretch are irreversible and history dependent, and are accompanied by fiber bundling) are by themselves not surprising, basically confirming prior experimental observations. But the computational modeling is novel and provides a useful framework for future studies of cell-matrix interactions. Until now, fibrous networks have mainly been modeled as permanently crosslinked networks that are elastic/reversible. The authors augmented a commonly used lattice model with load-induced bond breakage and peeling at branch points. They convincingly show that these minimal ingredients suffice to capture the effects of cell and external strains on collagen matrix structure/mechanics. A few minor questions/comments:

1) The authors point out that collagen bundles are present between "all cell pairs which are close enough". Please specify what is "close enough" and comment on the distance relative to cell size (several recent studies found cells exert forces across many times their own size in fibrous matrices) and relative to collagen network mesh size.

2) Data in Fig 1A-C: why are the cells round? Typically cells that are well-adhered to an extracellular matrix are elongated, often spindle-shaped, and they are typically modeled as force dipoles. Round cells usually are not well-adhered, which also negatively impacts cell traction forces.

3) Data in Fig 1D-F: the statistical significance of differences shown should be specified.

4) Fig 2: how would the predictions be impacted if the cells were to be modeled as contractile force dipoles, which is typically done in models of cell-matrix mechanobiology?

Reviewer #3 (Remarks to the Author):

This paper presents some results describing plasticity of cell remodeling of collagen gels, and proposes a computational model to explain the plasticity induced by cell-matrix interaction. There are some interesting results and insights provided by the experiments and modeling, and plasticity of collagen gels is certainly underappreciated in the field. However, the topics of stress dependent collagen gel plasticity, permanent or plastic remodeling of collagen gels by cells, and modeling of collagen gels based on bond breaking and reforming have all been reported in previous work by other groups. While these previous studies take different approaches, the work described in this manuscript does not provide a substantial enough advance relative to these other works to justify publication in Nature Communications. Further rationale for this is provided below. In addition to this central critique, there are a number of other critiques regarding the experiments and modeling that are noted below.

Major critiques:

1. Novelty of this manuscript with relation to other studies: A number of experimental papers have investigated the plasticity of collagen gels with relation to cells including:

-Petroll et al, Dynamic three-dimensional visualization of collagen matrix remodeling and cytoskeletal organization in living corneal fibroblasts, *Scanning*, 2004

-Mohammadi et al, Inelastic behaviour of collagen networks in cell-matrix interactions and mechanosensation *Interface*, 2015

-Nam et al, Viscoplasticity Enables Mechanical Remodeling of Matrix by Cells, *Biophys. J.*, 2016

In particular, the manuscript by Nam et al. describes stress-dependent plasticity in collagen gels, and establishes that plasticity mediates mechanical remodeling of the collagen gel by the cells.

These studies should be referenced in the introduction section of the manuscript, as the current wording makes it seem as if collagen gel plasticity has not previously been investigated.

Further, there are highly relevant modeling studies:

-Wolff et al, Inelastic mechanics of sticky biopolymer networks, *New J. Phys.* 2010.

-Wolff et al, Resolving the Stiffening-Softening Paradox in Cell Mechanics, *PLoS One.*, 2012

-Nam et al, Strain-enhanced stress relaxation impacts nonlinear elasticity in collagen gels, *PNAS*, 2016

-Semmrich et al, Glass transition and rheological redundancy in F-actin solutions, *PNAS*, 2007.

These works all address bond-breaking and reforming for biopolymer networks. These works should be cited, similarities/differences with the current model should be discussed, and justification for the difference in approaches should be provided. In particular, Nam et al., propose force-dependent unbinding of crosslinks between collagen based on experimental AFM data. In contrast, the authors here assumed strain/stress-independent bond breaking with a constant threshold and temporal exponential decay of bond reforming in their model. What is the rationale for choosing a different model for binding interactions between collagen? An experimental justification for strain/stress independent bond breaking with a threshold should be provided since this is different than what previous experimental results suggest.

2. The quantitative measure used to assess plasticity in experiments and modeling is not very precise and should be improved upon. The authors use the measure $\Delta F/F$ where ΔF is based on a quantification of fluorescence intensity within a manually drawn box between two cells and the average background fluorescence intensity far from the cells. This measure does not capture orientation of the fibers and has some intrinsic limitations. For example, the quantification is size dependent as if the box is drawn small enough such that it is spanned by one fiber, F_b and thus ΔF would be artificially large. There are numerous papers that have established approaches for identifying collagen fibers from fluorescence images (e.g. Stein, et al., *J. Microscopy*, 2008). Such approaches would provide a finer measure of how fiber thickness, density, and orientation changes, and could provide better feedback and assessment for the model.

3. I am concerned with the methodology used to measure the dependence of strain relaxation on dwell shown in figure 3D. The authors write "To measure the dependence of strain relaxation on dwell time, the dwell time starts from 1 minute and gradually increases to 20 minutes for each given sample. For each dwell time, we allow 15 minutes of relaxation before bringing back the sample to 20% strain. To account for the cumulative effect of the residual strains, we re-calibrate the system after each relaxation cycle by setting the previously relaxed configurations as strain zero states." For each dwell time experiment, the authors are inducing permanent structural changes in the network, and the collagen fiber orientation in particular, which may bias the subsequent experiments at a higher dwell times, making these results difficult to interpret. "re-calibrating" may not fully compensate for this bias. Experiments with different dwell times should be conducted on independent gels in independent experiments, or the ordering of the dwell time experiments should be randomized.

4. There are some assumptions with the model that should be clarified. First, the contraction factor β used for cell-contraction is stated as 40%-50% of cell-body. This seems like a large value for cell-body contraction. The authors should justify this. Second, in general, this does not seem to capture the way cells contract matrices. Particularly when cultured in fibrillar collagen gels, cells form adhesions to specific regions of the matrix, and then exert contractile forces on those adhesions. They do not contract gels uniformly. How would mimicking this contraction at discrete points change the results of this model?

5. The authors attributed plasticity of collagen networks only to cell traction forces. However, cells also secrete proteases to remodel collagen gels. Is there a contribution of proteases to the remodeling observed here, or can it all be attributed to traction forces as is claimed?

Minor points:

1. As this work is focused exclusively on collagen networks, the phrase "biopolymer networks" in the title should be replaced by "collagen networks".

2. The y-axis scale of strain in Fig.3C is so small, compared with the others (orders of magnitude smaller). Is this correct or this is a mistake?

3. What is the specific time used to determine the strain at infinity $\epsilon(\infty)$?

4. The definition/description of directional compliance should be included in the main manuscript. Also, how does the ellipse fit vary as a function of orientation to cell - for a given distance of say 5 μm , is it different between two cells versus 5 μm away from one cell in direction away from the other cell? This would be interesting for determining if the alteration in compliance between cells is just an additive effect of the impact of the two cells, or if there is some kind of synergy/feedback between the cells.

5. The title of supplementary information is different from the title of main manuscript. These should be matched.

6. Specific figures in the supplementary information should be cited in the manuscript at the appropriate points instead of a general "Supplementary Material" to make it easier for the reader to find the relevant figure.

7. In the abstract, the authors claim that "Our results reveal the profound impacts of cellular traction forces on their host ECM during development and cancer progression ...". This claim is far too broad as no studies have been conducted in a developmental model or an impact on cancer progression shown. The authors could replace the word reveal by suggest.

We thank the reviewers for the careful assessment of our manuscript. The reviewers' feedback has been very helpful in continuing to strengthen the manuscript. Below we give a point-by-point response to the reviewers' comments. **Changes to the manuscript are highlighted in red.**

-----Reviewer 1-----

In the paper titled "Stress-induced Plasticity of Dynamic Biopolymer Networks", Kim et al. have performed an experimental and computational study of Biopolymer networks found in the extra-cellular matrix. The experiments have been performed on Type 1 collagen gels whereas the simulations have been performed on the lattice based (2D and 3D) network models. Based on their experiments, they claim that the network is significantly remodeled by cellular traction forces. The remodeling is irreversible and can occur under the application of microscopic strains also. They have then developed a computational model, which takes into account the remodeling of network by including sliding and merging of network fibers. The model calculations are in qualitative agreement with experiments. The study highlights that traction forces from cell pairs can cause significant alignment of collagen fibers, resulting in fiber bundles. The authors emphasize that the traction forces cannot be treated as small perturbations as they can increase the fiber density by as much as 150%.

The findings of this study are interesting. That collagen matrix is irreversibly remodeled in presence of traction forces or macroscopic strains has been reported in previous experimental studies, the study of Kim et al. provides a computational approach which I believe is well suited to the study of remodeling of ECM. I find the article well written and interesting to warrant a publication in Nature communications. There are, however, certain points that must be addressed by the authors before I can recommend publication.

We thank the reviewer for taking time reading our manuscript, and we are glad the reviewer agrees that our results are interesting, well written, and warrant a publication in Nature Communications. We also thank the reviewer for the constructive comments that we have taken advantage of to improve the manuscript. Our point-by-point response to the reviewer's comments are detailed in below.

1. The remodeling here focuses on collagen networks where there is no generation of collagen fibers. The remodeling is purely due to the forces with no change in the total line density. Since remodeling of collagen networks, on a broader perspective, also takes into account generation of fibers, I suggest the authors to clearly mention the absence of generation of collagen fibers in the introduction to the article.

We have added in the introduction:

“The observed ECM remodeling is purely mechanical, without the creation or degradation of ECM fibers.”

2. The experimental results in Figure 1 are strongly indicative of the remodeling by traction

forces. However, the computational model details are not sufficient (including those in the Supplementary information) and occasionally miselading.

a. The authors mention time $T_{\text{sliding}} = 0.5$ min and $T_{\text{mergin}} = 25$ min. The only reason for this choice of parameters as mentioned by the authors in the Supplementary information is 'To reach good correspondence with experiments'. What is meant by this? In the computational model, where one minimizes the elastic energy (before sliding/merging events), using any appropriate algorithm (be it conjugate gradient), there is no notion of time. Even if the authors used steepest descent method and other standard assumptions (no hydrodynamics, over-damped), the natural time scale in the system will need to be expressed using viscosity of the underlying solvent and elastic forces in the network. Hence, I do not understand the meaning of T_{sliding} and T_{merging} numerical-values provided. Is it not so that in the simulations, the merging events are 5-times rarer in time than the sliding events (purely based on parameter choice of authors)? Please clarify the concept of time-evolution in the simulations.

Our simulation makes a standard quasi-static assumption, i.e. the viscoelastic relaxation is much faster than the plastic relaxation, which has been confirmed by the experiments. We have added discussion and references for this on page 3-4.

In our model, T_{sliding} is a sampling time. Every time step equals to T_{sliding} , we first minimize the elastic energy with a conjugate gradient algorithm, we then carry out the sliding events based on their computed probability. Therefore to make a comparison with experiments, we would like to keep T_{sliding} much smaller compared with the plastic relaxation time, but at the same time lower bonded by the viscoelastic relaxation time. Our experimentally measured viscoelastic relaxation time is approximately 30 sec, 10 times smaller than the plastic relaxation time scale. That is why we have chosen $T_{\text{sliding}} = 30$ sec.

Our choice of T_{merging} is also derived from the experiments. From Fig 1, we see the irreversibility of the system is significantly enhanced when the maturation time is increased from 0.2 hr to 1 hr, but less so after 1 hr. This observation suggests that T_{merging} should be chosen between 0.2 hr to 1 hr. We find that varying T_{merging} between 0.2 hr to 1 hr only subtly changes the simulation results. We therefore set $T_{\text{merging}} = 25$ min which fits experiments best. That being said, we agree that there may be more fundamental ways to derive T_{merging} , even though it turns out to be a non-sensitive parameter. One likely path would be to calculate the screened charge interactions between fibers. However, discussion on this topic goes beyond the scope of our current manuscript.

We have included these clarifications in the revised supplementary material (section S.2 – parameters)

Additionally, a less than 1min viscoelastic time scale was reported before. We therefore added the relevant references and discussion in the main text page 4-5 (“...In a real collagen matrix, ... Thus we ignore viscoelasticity in our modeling and only focus on plastic events including sliding and merging.”).

b. When sliding events are considered, the authors consider a frozen network configuration, i.e., all those fibers that can slide are allowed to do so, or one follows a priority list in order of most stressed to least stressed with network relaxation between every subsequent sliding event (a quasi-static approach)?

We consider a frozen network configuration, i.e. all rest lengths of bonds are updated at the same time. Typically there are multiple sliding events (especially in the very beginning of the relaxation) every T_{sliding} step in a 60 by 60 network. Despite that the priority list procedure might be more intuitive, we can't update one bond's rest length at each time and minimize the free energy accordingly due to computation limitation,

3. Assuming that the authors can explain the choice of $T_{\text{sliding}} = 0.5$ minutes, is this time scale independent of the initial stress/magnitude of traction forces?

Our current model assume T_{sliding} is independent of stress. On the other hand, we have considered an alternative formulism: T_{sliding} is still independent of stress, but the probability of each sliding event within T_{sliding} takes an exponential, rather than step function. In our manuscript, the probability is a step function. For a force on the crosslink node smaller than sliding threshold, $p=0$. Otherwise, $p=1$. To make it force dependent, we change p to $p=1-\exp(-\text{force}/\text{sliding threshold})$. In this case, the stronger forces cause higher sliding rate. With the same parameter set of the top curve in Fig3B (dwell time=20min), we redo the simulation with the force dependent sliding probability. As shown below, force dependence sliding gives very similar results to that of the original approach.

This additional simulation results is now included in the Supplementary Materials sections S.3

4. Page 3, 4th paragraph: "Affine to non-affine transformation through sliding events": Do the authors apply a step-shear to the system? Is the dwell time corresponding to the time for which the strain is kept constant? The authors say that for a dwell time of 20 minutes, they observe an affine to nonaffine transition that is attributed to plastic events. However, in the reference [21], the nonaffine transition has nothing to do with plastic events. There (in ref 21), the nonaffine transitions are found in a topologically fixed network.

We agree with the reviewer that the stamen of “affine-to-nonaffine transition” was misleading. So we have revised the manuscript as:

“Interestingly, we find that during the dwell time the stretching energy decays much faster than the bending energy, implying transitions from stretching-dominated states to bending-dominated states through sliding events.”

We have also added discussions of the background of the diluted triangular lattice model (its different elasticity regimes, nonlinear elasticity) on page 2 when we introduce the model (“... The linear and nonlinear elasticity... in good agreement with observations of various biopolymer gels”).

As for the initial strain, in experiments, the initial shear was applied at 1% per second, which was the fastest given the equipment and sample constraints to avoid overshoot and sample slippage. In the model, the initial strain was applied as a step function.

5. In Fig 3D, a double exponential function is used to fit the data. The time scales of the two exponentials are well separated. However in the plot 3D, the time (x axis) is shown in 100s of seconds such that one cannot possibly see the τ_v relaxation at all. The authors should zoom in the short time scale to show the fit to the τ_v .

We have included an inset to Fig. 3D to show the zoomed-in version at the initial 70 seconds.

6. I am really curious about why one should attempt an exponential (or double exponential) fit? Is there an obvious reason to assume exponential relaxation?

We found that exponential fit to be the most natural starting point, and indeed turned out to be working sufficiently well. This is not surprising given the fact that the dynamics is dominated by a finite number of time scales.

That being said, we are aware that some polymer networks, such as the cytoskeleton of live cells, follow power-law relaxation. There the non-exponential relaxation is partially contributed by the wide spectrum of time scales associated with many types of cytoskeleton-associated proteins.

A General remark:

The article is a good piece of work. I do not think that it needs to make speculations (last paragraph of discussions section) like "...will take advantage of collagen matrix as a mechanically programmable material which has excellent biocompatibility". I find it highly unnecessary but leave it to authors to decide whether they do anything about it. Similarly the very last sentence of the discussion section: "...cancer-metastasis, wound healing and embryo development." Is this really needed?

We appreciate the reviewer to share this view with us. However, we feel that these discussions will help put our work into context and suggest future applications. In fact, we are conducting researches along the line (particularly cancer metastasis) to substantiate these statements.

Citation error: 23 and 32 are repeated.

We thank the reviewer to point out this error. The citation is now fixed.

This manuscript reports a combined experimental and computational study of plasticity in fibrous extracellular matrix networks. This topic is currently gaining increasing interest in the context of cancer biology and regenerative medicine, where cell-driven remodeling of the matrix is a central phenomenon.

The main findings of the paper (large deformations applied by cells or external stretch are irreversible and history dependent, and are accompanied by fiber bundling) are by themselves not surprising, basically confirming prior experimental observations. But the computational modeling is novel and provides a useful framework for future studies of cell-matrix interactions. Until now, fibrous networks have mainly been modeled as permanently crosslinked networks that are elastic/reversible. The authors augmented a commonly used lattice model with load-induced bond breakage and peeling at branch points. They convincingly show that these minimal ingredients suffice to capture the effects of cell and external strains on collagen matrix structure/mechanics. A few minor questions/comments:

1) The authors point out that collagen bundles are present between "all cell pairs which are close enough". Please specify what is "close enough" and comment on the distance relative to cell size (several recent studies found cells exert forces across many times their own size in fibrous matrices) and relative to collagen network mesh size.

We thank the reviewer for the suggestion. **To make our statement precise, we have revised the original Fig. S6 (Fig. S2 in the revision) to reflect the signal-to-noise ratio (SNR) of the reflectance intensity. We have also added reference to the SI in the main text when we state 'the formation of collagen bundle is robust'.** Below we outline the main points of the revised supplementary figure S2 and supplementary methods section S1.

The background intensity follows a Gaussian distribution with standard deviation of δF . We distinguish bundles with $\Delta F > 3 \delta F$ (closed symbols of Fig. S2) with bundles with $\Delta F < 3 \delta F$ (open symbols of Fig. S2). If we consider criterion of a collagen bundle to be $\Delta F > 3 \delta F$, our data shows collagen bundle presents between cells pairs as far as 80 μm away. This is to be compared with the average cell radius (17 μm), and matrix pore size (3 μm). We have included these discussions in the revised supplementary material (supplementary methods section S1 – section cell-cell distances and cell morphology)

2) Data in Fig 1A-C: why are the cells round? Typically cells that are well-adhered to an extracellular matrix are elongated, often spindle-shaped, and they are typically modeled as force dipoles. Round cells usually are not well-adhered, which also negatively impacts cell traction forces.

These images were taken within the first few hours of seeding the cells. If allowed to culture a longer time (more than 8-10 hours), a distribution of cell morphology is typically. For the current study, the statistical distribution of the cell aspect ratios is shown in the supplementary material (Fig. S2).

We agree with the reviewer that usually elongated cells apply stronger contraction, making the formation of collagen bundle more pronounced. In the current study, focusing rounded cells helps reducing the complexity of modeling, and sheds light on the initial phase of bundle formation.

3) Data in Fig 1D-F: the statistical significance of differences shown should be specified.

We have included the ANOVA results in the revised graphics and caption of Fig. 1.

4) Fig 2: how would the predictions be impacted if the cells were to be modeled as contractile force dipoles, which is typically done in models of cell-matrix mechanobiology?

We thank the reviewer for pointing out this alternative model of cell traction force. We have performed new simulations in this case, and have included results in the supplementary material (Fig. S9 and also in below).

To mimic force dipoles, we modeled the cells as ellipse instead of circles. With the same parameter set of Fig 2C except that we change the cells from $r=6$ circles to $a=6, b=3$ ellipses. As shown below, the shape of curves in the alternate model differs somewhat from that in Fig 2C, but it is still clear that the plastic model predicts a higher concentration of fibers in region of bundles than the pure elastic model.

This paper presents some results describing plasticity of cell remodeling of collagen gels, and proposes a computational model to explain the plasticity induced by cell-matrix interaction. There are some interesting results and insights provided by the experiments and modeling, and plasticity of collagen gels is certainly underappreciated in the field. However, the topics of stress dependent collagen gel plasticity, permanent or plastic remodeling of collagen gels by cells, and modeling of collagen gels based on bond breaking and reforming have all been reported in previous work by other groups. While these previous studies take different approaches, the work described in this manuscript does not provide a substantial enough advance relative to these other works to justify publication in Nature Communications. Further rationale for this is provided below. In addition to this central critique, there are a number of other critiques regarding the experiments and modeling that are noted below.

Major critiques:

1. Novelty of this manuscript with relation to other studies: A number of experimental papers have investigated the plasticity of collagen gels with relation to cells including:

- Petroll et al, Dynamic three-dimensional visualization of collagen matrix remodeling and cytoskeletal organization in living corneal fibroblasts, Scanning, 2004
- Mohammadi et al, Inelastic behaviour of collagen networks in cell-matrix interactions and mechanosensation Interface, 2015
- Nam et al, Viscoplasticity Enables Mechanical Remodeling of Matrix by Cells, Biophys. J, 2016

In particular, the manuscript by Nam et al. describes stress-dependent plasticity in collagen gels, and establishes that plasticity mediates mechanical remodeling of the collagen gel by the cells. These studies should be referenced in the introduction section of the manuscript, as the current wording makes it seem as if collagen gel plasticity has not previously been investigated.

Further, there are highly relevant modeling studies:

- Wolff et al, Inelastic mechanics of sticky biopolymer networks, New J. Phys. 2010.
- Wolff et al, Resolving the Stiffening-Softening Paradox in Cell Mechanics, PLoS One., 2012
- Nam et al, Strain-enhanced stress relaxation impacts nonlinear elasticity in collagen gels, PNAS, 2016
- Semmrich et al, Glass transition and rheological redundancy in F-actin solutions, PNAS, 2007.

These works all address bond-breaking and reforming for biopolymer networks. These works should be cited, similarities/differences with the current model should be discussed, and justification for the difference in approaches should be provided. In particular, Nam et al., propose force-dependent unbinding of crosslinks between collagen based on experimental AFM data. In contrast, the authors here assumed strain/stress-independent bond breaking with a constant threshold and temporal exponential decay of bond reforming in their model. What is the rationale for choosing a different model for binding interactions between collagen? An experimental justification for strain/stress independent bond breaking with a threshold should be

provided since this is different than what previous experimental results suggest.

We thank the referee for the useful list of previous work related to our manuscript. We particularly agree that the works of Nam et al. (Biophys. J) and Nam et al. (PNAS) are very relevant to the present work. We did cite both papers in our original manuscript but we should have given more discussion. **We have fixed the references in our revised version and added some discussion of these two papers in the third and fourth paragraphs of the introduction. Here, in more detail, is our reaction to these works:**

The work of Nam et al. (Biophys. J.) indeed covers some of the same ground on cell-induced remodeling as we did. They emphasize the freezing in of fiber alignment, which we also consider. However, the multi-cellular effect of the formation of *permanent collagen bundles* is not in their work, and it, we believe, has important implications for cellular interactions in cancer invasion. The theoretical analysis in this paper is based on macroscopic phenomenology, and is superseded by the microscopic theory that we present.

The PNAS paper of Nam et al is similar in spirit to our theoretical work on the sliding model, but very different in detail. We model the disordered structure of the fiber networks using diluted lattices, which captures the complicated geometry of the actual biopolymer networks. The rich physics and phase diagrams of these fiber networks in both linear and nonlinear elasticity has been extensively discussed in the literature [Refs 20-23 in our manuscript]. The theoretical model in the listed papers by the referee all adopt single chain model and a mean field theory characterization of the network background. This is very risky for strongly disordered systems such as those we consider. We take full account of the disorder – this is new in our paper. It is considerably more challenging computationally, but essential for capturing complex geometry such as multiple cells interacting in the ECM instead of homogeneous strain.

The question about a fixed threshold for sliding rather than a more complex dependence on applied force has been answered above (Referee 1, point 3). There is very little difference in the results. We prefer the simpler theory.

2. The quantitative measure used to assess plasticity in experiments and modeling is not very precise and should be improved upon. The authors use the measure $\Delta F/F$ where ΔF is based on a quantification of fluorescence intensity within a manually drawn box between two cells and the average background fluorescence intensity far from the cells. This measure does not capture orientation of the fibers and has some intrinsic limitations. For example, the quantification is size dependent as if the box is drawn small enough such that it is spanned by one fiber, F_b and thus ΔF would be artificially large. There are numerous papers that have established approaches for identifying collagen fibers from fluorescence images (e.g. Stein, et al., J. Microscopy, 2008). Such approaches would provide a finer measure of how fiber thickness, density, and orientation changes, and could provide better feedback and assessment for the model.

We would like to thank the reviewer for pointing out alternative methods to characterize the collagen bundles.

In principle, we agree with the reviewer that $\Delta F/F$ depends on the sampling-window size. We have addressed this problem with two considerations: first, all our sampling windows are at least 5 times greater than the pore size (Fig. S1B); second, we calculate both the mean and fluctuation of the background using the same (finite) window size (Fig. S1C), and making sure the bundle is statistically distinguishable from the background (Fig. S1D).

On the other hand, we avoid using fiber-tracing methods because these algorithms typically require many empirical input parameters, and are unnecessarily complicated for our simple purpose of quantifying fiber densification. For instance, the algorithm described in Stein, et al., *J. Microscopy*, 2008 (which was written by one of the authors in the current manuscript) takes 17 parameters, and does not account for fiber thickness.

Finally, we thank the reviewer for suggesting measures of collagen fiber alignment. To this end, and to avoid the potential problems of fiber-tracing mentioned in the above, we apply a convolution-based method we have developed in the past. Briefly, we compute the coarse-grained nematic field, which shows the level and direction of average orientations in the confocal images. **We have included the results in the revised Fig. S3, which confirm that collagen bundles are aligned and densified regions caused by cell traction forces.**

3. I am concerned with the methodology used to measure the dependence of strain relaxation on dwell shown in figure 3D. The authors write “To measure the dependence of strain relaxation on dwell time, the dwell time starts from 1 minute and gradually increases to 20 minutes for each given sample. For each dwell time, we allow 15 minutes of relaxation before bringing back the sample to 20% strain. To account for the cumulative effect of the residual strains, we re-calibrate the system after each relaxation cycle by setting the previously relaxed configurations as strain zero states.” For each dwell time experiment, the authors are inducing permanent structural changes in the network, and the collagen fiber orientation in particular, which may bias the subsequent experiments at a higher dwell times, making these results difficult to interpret. “re-calibrating” may not fully compensate for this bias. Experiments with different dwell times should be conducted on independent gels in independent experiments, or the ordering of the dwell time experiments should be randomized.

We thank the reviewer for raising this very important question regarding experiment details. The reason we choose to use re-calibrating on single sample is to avoid sample-to-sample variations, which is known to be large for collagen matrices. Following the reviewer’s suggestion, we have performed new experiments where each sample is only used for one dwell time. The results are consistent with our original experiments (re-calibrating), with notable sample-to-sample variations. **These new results are included in the revised supplementary Fig. S15 and supplementary methods section S3.**

4. There are some assumptions with the model that should be clarified. First, the contraction factor β used for cell-contraction is stated as 40%-50% of cell-body. This seems like a large value for cell-body contraction. The authors should justify this. Second, in general, this does not

seem to capture the way cells contract matrices. Particularly when cultured in fibrillar collagen gels, cells form adhesions to specific regions of the matrix, and then exert contractile forces on those adhesions. They do not contract gels uniformly. How would mimicking this contraction at discrete points change the results of this model?

We thank the reviewer to point out the apparently large contraction factor. Experimentally measured cell body compressive strain is typically 10%-30% (M.S.Hall et al PNAS 113(49) 14043 2016), which is smaller compared with the β value we used. We think the discrepancy may come from two sources. One is dimensionality. The experimentally observed fiber densification has contributions from all three dimensions. The model, on the other hand, is 2D. To account for the same level of fiber densification, we expect a larger β has to be used. The other reason is that cells do not have to change volume in order to apply traction force. It is well known from 3D traction force microscopy that cells exert strong forces that are tangential to the cell membrane. The tangential force can be generated by the relative shear between cells and the ECM. In our 2D model, however, traction force can only be generated by shrinking the area of the model cell. **We have clarified this in the revised supplementary methods section S2 - parameters.**

We also thank the reviewer to point out that the model may be improved to account for the discrete sites of adhesions. On the other hand, two experimental observations appear to favor the uniform contraction: (1) the deformation field is mostly uniform (Fig. 1 main text), (2) adhesion complexes (**visualized via vinculin staining, see revised supplementary material Fig. S2** and also attached below) either diffusively or punctually cover most of the cell membrane. That being said, we can easily adjust our model to account for discrete sites of force generation. As a particular example, the case where each cell is a force dipole can be effectively represented by elliptical cell shapes. **We have obtained new simulation results for elliptical cells as shown in the revised supplementary material Fig. S9.**

5. The authors attributed plasticity of collagen networks only to cell traction forces. However, cells also secrete proteases to remodel collagen gels. Is there a contribution of proteases to the

remodeling observed here, or can it all be attributed to traction forces as is claimed?

We have confirmed the mechanical origin of the collagen bundles using two different methods. First, by using a microstretcher device we have shown that purely mechanical deformation can lead to plastic deformation (Fig. S5). Second, we have performed new experiments where we inhibit the protease activity but still see formation of collagen bundles. **The result is included in the revised supplementary material Fig. S6, and attached below.**

Minor points:

1. As this work is focused exclusively on collagen networks, the phrase “biopolymer networks” in the title should be replaced by “collagen networks”.

We have confirmed that another example of biopolymer network, fibrin gel, also demonstrates similar plasticity as shown below: the long relaxation time scale τ_p and residual strain both increase with dwell time. This provides additional evidence that plasticity is a general feature of biopolymer networks and we therefore decide to keep our original title. **The new results of fibrin gel are included in the revised supplementary figure S16.**

2. The y-axis scale of strain in Fig.3C is so small, compared with the others (orders of magnitude smaller). Is this correct or this is a mistake?

The y-axis of Fig. 3C was mislabeled, we have fixed the labeling in the revision.

3. What is the specific time used to determine the strain at infinity $\epsilon(\infty)$?

We have used 15min to determine $\epsilon(\infty)$. Because the longest relaxation time is about 5 min, there is little change of the strain after 15min. The figure in below shows the change of strains between 15 and 25 minutes for five different samples each hold with different dwell times. (A: the strain relaxation dynamics. B: the residual strain measured at 15 and 25 minutes)

4. The definition/description of directional compliance should be included in the main manuscript. Also, how does the ellipse fit vary as a function of orientation to cell - for a given distance of say 5 μm , is it different between two cells versus 5 μm away from one cell in direction away from the other cell? This would be interesting for determining if the alteration in compliance between cells is just an additive effect of the impact of the two cells, or if there is some kind of synergy/feedback between the cells.

We have included the definition of directional compliance in the main text, immediately after we mention it for the first time in the text.

We thank the reviewer for pointing out that additional information might be extracted from the micromechanics measurements. We indeed explored the data along the suggestion of the reviewer. However, other than only a few instances, we do not have enough samples to draw a solid conclusion. We plan on expanding the investigation of mechanical feedback in the future.

5. The title of supplementary information is different from the title of main manuscript. These should be matched.

We have fixed this error.

6. Specific figures in the supplementary information should be cited in the manuscript at the appropriate points instead of a general "Supplementary Material" to make it easier for the reader to find the relevant figure.

We have revised the main text according to the reviewer's suggestion.

7. In the abstract, the authors claim that "Our results reveal the profound impacts of cellular traction forces on their host ECM during development and cancer progression ...". This claim is far too broad as no studies have been conducted in a developmental model or an impact on cancer progression shown. The authors could replace the word reveal by suggest.

We agree with the reviewer and have made the change in the revision.

Reviewers' comments:

Reviewer #1 (Remarks to the Author):

The authors have satisfactorily answered the questions and made appropriate changes to the manuscript. I recommend publishing of the revised article in Nature communications.

Reviewer #2 (Remarks to the Author):

The authors mostly dealt satisfactorily with all the comments and queries from the reviewers. The additional experimental data are a useful addition to the paper. It would be useful if some details were added: in particular about the rate of initial shear application (reviewer 1, point 3), and the details of the image analysis used to determine fiber orientations (reviewer 3, point 2).

Reviewer #3 (Remarks to the Author):

In the revised manuscript the authors have been partially responsive to some of the concerns raised in my previous review. The authors have clarified the advance of the manuscript, included additional experiments and computational simulations that address some of the questions, and improved the manuscript. While this is appreciated, and while I think the manuscript is close to being scientifically suitable for publication, this reviewer's opinion is that the manuscript still does not reach the level of impact typical for Nature Communications.

In their response, the authors clarify that the main contribution of this manuscript is in the development of a more complex computational model of collagen fiber network remodeling by cells that captures the "disorder" of collagen networks and the geometry of irreversible remodeling by cells, which is not captured in previous mean-field theory models. They then validate by examining collagen fiber remodeling between cell pairs in collagen gels. While I agree with that this is an advance beyond previous models, there are a number of limitations in the model, which I mostly brought up in the previous round of review, that diminish this reviewer's enthusiasm for its use and impact on the field. First, the authors use a 2D diluted triangular lattice model, which while simple and cost-effective, also does not capture the full complexity and disorder of actual biopolymer networks, which are 3D and have fibers whose contours may deviate in a meaningful way from the lattice. Second, whether the model can accurately predict changes in collagen fiber orientation is not clear. Changes in density and collagen orientation both underlie plasticity. This model is only used to predict changes in density. A key question is whether it can also capture changes in collagen fiber orientations. Third, the assumption used by the model of 50% uniform contraction of a circle (or ellipse) do not capture key aspects of cell-collagen network interactions. There is not any evidence, to this reviewer's knowledge, that cells themselves reduce their volume substantially during contraction of collagen gels. The Hall, et al., paper cited by the authors does show compressive strains of at a maximum ~12% (the data point of 30% strain cited by the authors in their response appears to be an outlier data point on panel 3A), but this is only along the major axis of the body, and is not uniform. Even if we assume the contraction can be uniform, there is a large difference between 12% strain and 50% strain. The authors suggest that this discrepancy in traction is due to either the model being 2D or that the model cannot capture shearing forces from cells. Either of these possibilities highlight a limitation of this model. Further, cell-collagen network adhesions occur at discrete points, given the microporous nature of the collagen network. While the authors performed a simulation in the response that involves the uniform contraction of an ellipse, uniform contraction of an ellipse does not capture contraction at discrete points. While the authors could find parameters such that the model results matched the experimental results, the rationale for these parameters does not seem to be strongly guided by experimental data.

Additional comments:

1. There are other important manuscripts that describe plasticity in cell-collagen network interactions that should be cited. These were noted in my last review: Petroll et al, Dynamic three-dimensional visualization of collagen matrix remodeling and cytoskeletal organization in living corneal fibroblasts, *Scanning*, 2004 and Mohammadi et al, Inelastic behaviour of collagen networks in cell-matrix interactions and mechanosensation, *J.R. Soc. Interface*, 2015. The modeling work of Wolff, et al., and Semmrich, et al, that consider dynamic bonds in biopolymer networks I noted previously could be cited, though if the scope of the manuscript is restricted to collagen networks, then this is not strictly necessary.

2. The authors have justified their use of the term "Biopolymer networks" with the addition of one panel of mechanical measurements on fibrin that is included in the supplementary information, which is not mentioned in the manuscript. Biopolymer networks refer to a wide class of molecules including intracellular polymers such as actin, neurofilaments, vimentin which have different molecular structures, architectures, and types of crosslinking interactions. The use of the term "biopolymer networks" in the title is misleading and overstates the work done in this paper. The authors should use "collagen networks" instead of "biopolymer networks" in the title.

3. I am concerned that the time used to characterize strain at infinity is only 15 minutes, as the strain still seems to be decreasing at that point. The authors should use a longer time if they want to describe this as strain at infinity. If the author presented the data with the x-axis on a log scale, as is typical for these types of experiments, it would be clearer what an appropriate timescale should be. If the authors decide to continue using 15 minutes, they should not call this strain at infinity, as that could be misleading.

We thank the reviewers for the careful assessment of our manuscript. While both reviewer 1 and reviewer 2 recommended publication of our work, reviewer 3 argued against it. However, while we are willing to and have made additional changes to the manuscript, we find reviewer 3's concerns are mostly about fine technical details, but not about the novelty and impact of our study. Below we give a point-by-point response to the reviewers' comments. **Changes to the manuscript are highlighted in red in this letter and the revised manuscript.**

-----Reviewer 1-----

The authors have satisfactorily answered the questions and made appropriate changes to the manuscript. I recommend publishing of the revised article in Nature communications.

We thank the reviewer for recommending publication of our manuscript. Previous comments from reviewer 1 have significantly helped us to strengthen our manuscript.

-----Reviewer 2-----

The authors mostly dealt satisfactorily with all the comments and queries from the reviewers. The additional experimental data are a useful addition to the paper. It would be useful if some details were added: in particular about the rate of initial shear application (reviewer 1, point 3), and the details of the image analysis used to determine fiber orientations (reviewer 3, point 2).

We thank the reviewer for recommending publication of our work, and appreciate the reviewer's previous comments that helped us to clarify important points in the previous revision.

The rate of initial shear application was 1% per second, and we have already included this information in the method section of the previous revision (section Methods-C: Measurement and fitting of the bulk relaxation kinetics).

The brief step-by-step image analysis for fiber orientation has been included in the caption of Supplementary Figure S3. The image analysis was described in great detail by us previously. And we ensure this reference is included in both the supplementary text (section: Fiber alignment in the collagen bundles) and the caption of Supplementary Figure S3.

-----Reviewer 3-----

In the revised manuscript the authors have been partially responsive to some of the concerns raised in my previous review. The authors have clarified the advance of the manuscript, included additional experiments and computational simulations that address some of the questions, and improved the manuscript. While this is appreciated, and while I think the manuscript is close to being scientifically suitable for publication, this reviewer's opinion is that the manuscript still does not reach the level of impact typical for Nature Communications.

We appreciate reviewer 3 spending time reading our manuscript again. The previous comments from reviewer 3 had significantly reshaped how we deliver our results. For that we are very thankful. However, we respectfully disagree with reviewer 3's overall assessment of our work, which is in clear contrast with those of the other two reviewers. In fact, there is no indication of why he/she believes that our paper does not have the needed impact, and in fact the residual concerns are mostly minor and technical. Some of these have already been addressed in the previous revision (and are further clarified in the current revision) and some are new. Here we present a point-by-point response to these technical points.

In their response, the authors clarify that the main contribution of this manuscript is in the development of a more complex computational model of collagen fiber network remodeling by cells that captures the “disorder” of collagen networks and the geometry of irreversible remodeling by cells, which is not captured in previous mean-field theory models. They then validate by examining collagen fiber remodeling between cell pairs in collagen gels. While I agree with that this is an advance beyond previous models, there are a number of limitations in the model, which I mostly brought up in the previous round of review, that diminish this reviewer's enthusiasm for its use and impact on the field.

We do not claim that our model is realistic in every detail, but rather, we want to highlight the most important features that explain the experimental data. These features are: the disordered nature of collagen matrix, the dynamic bonds between collagen fibers, and plastic remodeling by forces either generated by the cells or applied macroscopically. To our knowledge, and as pointed out by reviewer 1 and reviewer 2 previously, such a model is new to the field, and will be of broad interest to larger scientific community. In fact, we think the simplifying assumptions we made in our model lead to significant computational advantages that will only broaden the use of our model by other researchers.

First, the authors use a 2D diluted triangular lattice model, which while simple and cost-effective, also does not capture the full complexity and disorder of actual biopolymer networks, which are 3D and have fibers whose contours may deviate in a meaningful way from the lattice.

We have expanded discussion of this point in the revised manuscript to introduce existing literature that compared 2D lattice, off-lattice and 3D models. These results demonstrate that the main features of the elastic behavior are faithfully captured by the 2D triangular lattice model.

The new paragraph reads:

The relationship between this two-dimensional model and a real three-dimensional biopolymer networks has been examined in several recent papers. In particular, in [34] lattice models are compared to off-lattice versions. The result of all these studies is that the main features of the elastic behavior are faithfully captured, though there may be differences in details, particularly in the non-linear regime.

Second, whether the model can accurately predict changes in collagen fiber orientation is not clear. Changes in density and collagen orientation both underlie plasticity. This model is only used to predict changes in density. A key question is whether it can also capture changes in collagen fiber orientations.

The change of collagen fiber orientation is indeed predicted by the model as we have reported in the previous supplementary material (Fig. S8 B, Fig. S8D).

Third, the assumption used by the model of 50% uniform contraction of a circle (or ellipse) do not capture key aspects of cell-collagen network interactions. There is not any evidence, to this reviewer's knowledge, that cells themselves reduce their volume substantially during contraction of collagen gels. The Hall, et al., paper cited by the authors does show compressive strains of at a maximum ~12% (the data point of 30% strain cited by the authors in their response appears to be an outlier data point on panel 3A), but this is only along the major axis of the body, and is not uniform. Even if we assume the contraction can be uniform, there is a large difference between 12% strain and 50% strain. The authors suggest that this discrepancy in traction is due to either the model being 2D or that the model cannot capture shearing forces from cells. Either of these possibilities highlight a limitation of this model. Further, cell-collagen network adhesions occur at discrete points, given the microporous nature of the collagen network. While the authors performed a simulation in the response that involves the uniform contraction of an ellipse, uniform contraction of an ellipse does not capture contraction at discrete points. While the authors could find parameters such that the model results matched the experimental results, the rationale for these parameters does not seem to be strongly guided by experimental data.

We thank the reviewer's critical reading of our manuscript. We want to make clear that we are not modeling the plasma membrane, but rather the deformation of the extracellular matrix. Matrix close to the cell may deform significantly without apparent changing the cell volume, as the reviewer noted before. The contractility β is a measure of cell traction force in our model. Also while we agree that cell-ECM adhesions are discrete in nature, the distance between them is much smaller as compared to the distance we evaluate the deformation therefore can be treated as uniform and continuous. Indeed, we have experimentally shown that the adhesions are approximately uniform and continuous at the cell-ECM interface (Supplementary Fig. S2B inset).

We have added a new paragraph in the manuscript to clarify these points. The new paragraph reads:

Note that our computational cells contract by the large fraction β , whereas in our experiment (and in many others) the observed cell area does not change much in the process. This paradox is only apparent: we are not modeling the plasma membrane of the cell (which determines the observed area) but rather the motion of the points where the cell is attached to the surface. These points do contract strongly due to the action of motor proteins inside the cells. Further, we use a continuous disk of attachment rather than discrete points. This is also not a real problem: we are interested in deformations of the matrix at distances large compared to the spacing between adhesions. There is no difference between discrete and continuous adhesion in this regime. Another indication of this fact is that our results do not change much for elongated cells (Supplementary Fig. S9).

Additional comments:

1. There are other important manuscripts that describe plasticity in cell-collagen network interactions that should be cited. These were noted in my last review: Petroll et al, Dynamic three-dimensional visualization of collagen matrix remodeling and cytoskeletal organization in living corneal fibroblasts, Scanning, 2004 and Mohammadi et al, Inelastic behaviour of collagen networks in cell-matrix interactions and mechanosensation, J.R. Soc. Interface, 2015. The modeling work of Wolff, et al., and Semmrich, et al, that consider dynamic bonds in biopolymer networks I noted previously could be cited, though if the scope of the manuscript is restricted to collagen networks, then this is not strictly necessary.

We thank the reviewer for these references. Indeed our scope is not restricted to collagen networks, **therefore we have included the new references.**

2. The authors have justified their use of the term “Biopolymer networks” with the addition of one panel of mechanical measurements on fibrin that is included in the supplementary information, which is not mentioned in the manuscript. Biopolymer networks refer to a wide class of molecules including intracellular polymers such as actin, neurofilaments, vimentin which have different molecular structures, architectures, and types of crosslinking interactions. The use of the term “biopolymer networks” in the title is misleading and overstates the work done in this paper. The authors should use “collagen networks” instead of “biopolymer networks” in the title.

We have changed our title accordingly.

3. I am concerned that the time used to characterize strain at infinity is only 15 minutes, as the strain still seems to be decreasing at that point. The authors should use a longer time if they want to describe this as strain at infinity. If the author presented the data with the x-axis on a log scale, as is typical for these types of experiments, it would be clearer what an appropriate timescale should be. If the authors decide to continue using 15 minutes, they should not call this strain at infinity, as that could be misleading.

We have addressed this point in the previous response letter and revision (reviewer 3 minor point 3). We now make clear in the main text (**caption of Fig 3C**) that the infinite strain was approximated by experimentally measured strain at 15 min, so to avoid any misleading. On a related note, we did perform longer relaxation measurements, as shown in the Supplementary Fig. S16.

REVIEWERS' COMMENTS:

Reviewer #3 (Remarks to the Author):

I thank the authors for their responses and the additional changes made to the manuscript. These address my concerns.